# Symmetry Breaking during Cell Movement in the Context of Excitability, Kinetic Fine-Tuning and Memory of Pseudopod Formation

**DOI:** 10.3390/cells9081809

**Published:** 2020-07-30

**Authors:** Peter J.M. van Haastert

**Affiliations:** Department of Cell Biochemistry, University of Groningen, Nijenborgh 7, 9747 AG Groningen, The Netherlands; p.j.m.van.haastert@rug.nl; Tel.: +31-50-3634209

**Keywords:** pseudopod, Ras activation, cytoskeleton, *Dictyostelium*, chemotaxis, neutrophils

## Abstract

The path of moving eukaryotic cells depends on the kinetics and direction of extending pseudopods. Amoeboid cells constantly change their shape with pseudopods extending in different directions. Detailed analysis has revealed that time, place and direction of pseudopod extension are not random, but highly ordered with strong prevalence for only one extending pseudopod, with defined life-times, and with reoccurring events in time and space indicative of memory. Important components are Ras activation and the formation of branched F-actin in the extending pseudopod and inhibition of pseudopod formation in the contractile cortex of parallel F-actin/myosin. In biology, order very often comes with symmetry. In this essay, I discuss cell movement and the dynamics of pseudopod extension from the perspective of symmetry and symmetry changes of Ras activation and the formation of branched F-actin in the extending pseudopod. Combining symmetry of Ras activation with kinetics and memory of pseudopod extension results in a refined model of amoeboid movement that appears to be largely conserved in the fast moving *Dictyostelium* and neutrophils, the slow moving mesenchymal stem cells and the fungus *B.d. chytrid*.

## 1. Introduction

Amoeboid cells move by extending protrusions [1]. The shape of amoeboid cells is very flexible with frequent changes of extensions and directions of movement. These cells have an asymmetric appearance with seemingly infinite ways to construct the amoeboid body. However, careful analysis of how cells extend protrusions and the resulting trajectories have shown that movement is not random [2,3]. Many amoeboid cells have the tendency to persist in the direction of movement. Cells have a front and a rear that, although flexible, are relatively stable on a minute time scale. Such cells have a longitudinal axis with (imperfect) symmetry. Here symmetry is used to describe order; the infinite ways to construct an asymmetrical amoeboid body are restricted by the extending pseudopod. The extending pseudopod brings order and breaks the symmetry of the cell. This symmetry breaking is not static, such as the creation of the stable bilateral body plan during animal development, but is very dynamic, since cells extend a new pseudopod every ~15 s.

After working for many years on the biochemistry of cell movement and chemotaxis, and after a period serving the university as dean of education, I returned to science performing experiments on what seemed to me the fundament of cell movement: how do cells extend pseudopods and what is the underlying mechanism? In this essay, I combine the results of these recent studies [4,5,6,7] to present my view on the mechanism of pseudopod extension, discussed from the perspective of symmetry breaking. Pseudopod extension is regulated by many signaling molecules, especially during directed cell movement guided by chemoattractant. In *Dictyostelium*, members of the Ras family of GTPases that are detected with RBD-Raf-GFP appear to play an important role, being an upstream signal with strong local activation, both during cell movement in chemotactic gradients and in buffer [5,8]. Here, I use active Ras-GTP as a molecular anchor for symmetry. In the next paragraphs, I first give some background on symmetry forms, and their changes in time and space, on the cytoskeleton, and on the coupling of excitable Ras-GTP activation and the cytoskeleton. The core of the essay is the discussions on symmetry during cell movement, that starts with uniform Ras activation in a round cell, to which different components of the cytoskeleton are added based on experiments, thereby obtaining the complex symmetry form of polarized cells. The resulting model of amoeboid movement reveals a very rich pallet of regulatory mechanisms that provide amoeboid cells with complex symmetry forms, memory and refined kinetics of cell movement. Interestingly, the fundaments of symmetry for cell movement uncovered in *Dictyostelium* are also detectable in very different cells, such as the fast moving neutrophils, the slow moving mesenchymal stem cells or the fungus *B.d. chytrid*.

## 2. Fundaments of Symmetry and Symmetry Breaking in Cell Movement

Symmetry means the existence of different viewpoints from which the system appears the same. Symmetry breaking is the process by which the number of these viewpoints (the order of symmetry) is reduced, to generate a more ordered, structured and improbable state [9]. The term order is easily confusing, because it has opposite meaning in thermodynamics and symmetry. For instance, as the symmetry breaks from rotation symmetry to reflection symmetry, the system becomes more ordered thermodynamically, while the order of symmetry (the number of viewpoints) decreases. To avoid this confusion, I will use the term complexity to indicate that both the structure and the symmetry become more complex. In biology, symmetry is not precise as in physics. Furthermore, symmetry is sometimes permanent, such as the shape of a sea star, but in cell movement and many other processes in biology, symmetry is dynamic. When symmetry is disturbed, the system may stay disturbed, it may return to the original symmetry state, or it may adopt a more complex symmetry, depending on the underlying mechanisms. To understand the role of dynamic symmetry in biology, it is essential to understand these underlying mechanisms. Some of these aspects of symmetry are illustrated in Figure 1a,b using objects that are constructed with elements from my greenhouse. Firstly, symmetry is not exact and may depend on the point of interest. The object of Figure 1a clearly appears to have 5-fold rotational symmetry: the five elements are approximately equal in size and arranged at approximately equal angles. However, in its details the elements have different width, color and curvature, and the five elements are oriented differently with curvature to the right in three elements and to the left in two elements. Therefore, the object is symmetric in some studies e.g., how local activators and inhibitors lead to rotational symmetry in a sea star, but the object is non-symmetric in other studies e.g., how elements in the object can get a left- or right-handed curvature such as in the rotation of runner beans and French beans. Secondly, symmetry is often dynamic. When one element disappears (Figure 1b) the object gets imperfect 5-fold rotational symmetry (one element has zero size) or imperfect 4-fold rotational symmetry (all four angles are different from 90 degrees). Interestingly the object gets reflection symmetry that is nearly perfect. If the underlying mechanism is rotational symmetry, the object may recover 5-fold symmetry requiring the growth of a new element, or 4-fold symmetry requiring that the position of the four elements adopt 90 degrees angles. However, if the system has the ability to form a polarity axis, the reflection symmetry may be enforced by specific growth of some elements and shrinkage of other elements. Thus, the disturbed structure dynamically changes to a regular structure with a symmetry that depends on the underlying mechanisms.

These aspects of symmetry are illustrated in Figure 1c,d for a *Dictyostelium* cell expressing a sensor for active Ras-GTP and extending pseudopods [4]. Depending on the viewpoint (pseudopod extension or Ras-GTP patches; Figure 1c), the same object may have different symmetry forms. The cell has multiple patches of Ras-GTP that are distributed nearly evenly around the cell: Ras-GTP patches have rotational symmetry. Often the cell is somewhat elongated with only one extending pseudopod: movement has reflection symmetry. Shape and movement has fewer viewpoints of symmetry than the Ras-GTP patches, therefore shape is a more complex symmetry state than Ras-GTP patches. Furthermore, the more complex symmetry state may depend on a symmetry state with less complexity. If this cell is followed in time using the kymograph of Figure 1d, it appears that the 53 Ras-GTP patches are dynamic with a life time of about 24 s; on average the cell has 3 to 4 patches, and when a Ras-GTP patch disappears a new Ras-GTP patch is initiated and the cell keeps rotational symmetry of Ras-GTP patches. This cell extends only 14 pseudopods with a life time of about 15 s; usually a cell extends only one pseudopod at the same time, far less than the 3 to 4 Ras-GTP patches. Importantly, when a new pseudopod is made, it always starts at a place of a Ras-GTP patch, and nearly always at the Ras-GTP patch with the highest intensity. Thus, although the shape of the cell has reflection symmetry, the underlying mechanism is rotational symmetry of Ras-GTP patches. Consequently, the pseudopods start at different sides of the cell, and these briefly starved cell moves in nearly random direction. When *Dictyostelium* cells are starved for prolonged periods, they become polarized in shape and Ras activation: Cells still have multiple Ras-GTP patches, but the intensity is much higher in the patch at the current front of the cell: Ras-GTP patches have reflection symmetry (see also below in Figure 2). New pseudopods are still formed at the strongest Ras-GTP patch and therefore all pseudopods start in the front near the existing pseudopod. Consequently, the cell moves with persistence. The difference in persistent starved cells and random movement of non-starved cells is not the shape of the cell, but the underlying symmetry form of the activating Ras-GTP patches.

## 3. The Cytoskeleton of Moving Cells

The two major parts of the cytoskeleton of moving cells are branched F-actin filaments in the extending protrusion (bF-actin) and parallel F-actin filaments (pF-actin) in the contractile cortex [1,10,11,12]. Cryogenic Electron Microscopy has revealed that an emerging pseudopod contains in the order of 4000 branched actin filaments directing towards the membrane of the extending pseudopod [6,13]. The bF-actin is regulated by nucleation of Arp2/3 branch points that generate new filament ends from which polymerization can occur. Extension of the pseudopod will continue as long as new branches and filaments are formed, and as long as sufficient place is available for sufficient time between the end points of these filaments and the plasma membrane. At some moment, the balance between elongation and counter forces reverses, further polymerization comes to an arrest, and the pseudopod stops. Nucleation of the Arp2/3 is induced by Scar, a complex of five proteins that is regulated by Rac-GTP and its upstream activator Ras-GTP [14].

The contractile cortex is a thin 100 nm thick layer under the plasma membrane. It consists of long parallel F-actin filaments, cross linkers such as α-actinin, membrane anchor proteins, and myosin II filaments that can provide contractile force [12,15,16,17]. In regions with a strong contractile cortex, it is difficult to generate a new protrusion of branched F-actin [18,19,20]. In this respect, bF-actin in the extending pseudopod and pF-actin in the contractile cortex have opposite function, but both are detected with many F-actin sensors such as LimE-GFP and lifeact-GFP [21]. Myosin II filaments disassemble by phosphorylation of the heavy chain by myosin heavy chain kinase (MHCK) [22,23]. Disassembly of myosin filaments weakens the contractile cortex thereby increasing the probability that branched F-actin can be formed inducing a new pseudopod. Polymerization of pF-actin in the cortex is regulated by formins, ring-like structures that are activated by the small GTPase RacE. Mutants lacking three formins (*forAEH*-null) or RacE have a very weak contractile cortex and extend multiple pseudopods [16].

## 4. Coupled Excitable Ras/bF-Actin

Many observations suggest that F-actin forms an excitable medium [4,24,25,26,27,28]. Small fluctuations are damped, but fluctuations above a threshold lead to strong amplification with the characteristics of standing waves. The small GTPase Ras can exist in the inactive Ras-GDP and active Ras-GTP form. Sensitive sensors for the active Ras-GTP form suggest that, in the absence of F-actin, activation of Ras is also excitable leading to multiple patches of Ras-GTP with a diameter of about 3 µm and a life time of about 16 s [4]. Interestingly, excitable Ras and excitable F-actin are coupled: Ras induces F-actin, and F-actin further increases Ras activation. Cells usually have only one extending pseudopod enriched with F-actin, but multiple Ras-GTP patches. Therefore, when a new pseudopod is formed it is induced at the position of the strongest Ras-GTP patch, and upon induction of F-actin in this emerging pseudopod Ras is further activated to a very strong Ras-GTP patch with increased size (from 3 to 7 µm) and increased life-time (from 16 to 42 s). In a coupled excitable system, fluctuations above a threshold of either component can lead to the excitation of both. Thus a new pseudopod can start with an increase of F-actin in a patch of Ras-GTP (that further activates Ras-GTP/F-actin) or with an increase of Ras-GTP (that induces F-actin and further activation of Ras-GTP). Indeed, detailed analysis revealed that new pseudopods in the front of the cell (that contains bF-actin and lacks a contractile cortex) start with an increase of F-actin (presumably bF-actin), while a new pseudopod that occasionally is formed in the contractile cortex can only start with a strong patch of Ras-GTP that must first locally weaken the contractile cortex by disassembly of myosin filaments, followed by induction of branched F-actin [4,6,7].

## 5. Symmetry of Moving Cells

The symmetry of Ras patches, F-actin and cell shape has been analyzed in wild-type cells and a series of mutants with defects in components of the cytoskeleton or its regulation [5]. The role of F-actin (both bF-actin in the front and pF-actin in the contractile cortex) was studied with the use of the inhibitor Latrunculin A (LatA) and the role of myosin II in the contractile cortex in the rear of the cell with a null mutant. The border between bF-actin in the front and pF-actin in the rear is enriched with IQGap2/cortexillin/myosin II [29]. From this information a hierarchy of symmetry forms can be deduced (Figure 2).

Cells lacking functional F-actin, myosin and IQGAP (*myoII*-null + LatA) are round and have uniform low levels or Ras-GTP; i.e., they are in a low basal state. Incorporating F-actin (*iqgap2*-null without latA) induces some deformation in an otherwise rather round cell; importantly Ras-GTP levels are strongly elevated but still nearly uniform. In both mutant situations the cell has indefinite rotational symmetry. Cells that do have IQGap2 but no functional F-actin and myosin filaments (early wild-type + LatA) are also round, but now Ras-GTP is located in patches. These patches are rather uniform in size (3 µm) and life-time (16 s); most cells have 3 to 5 Ras-GTP patches that are approximately equally distributed over the circumference of the cell [4]. Thus, these cells have 3–5 fold rotational symmetry, and symmetry is due to inhibition of Ras-GTP formation in between the patches by IQGap2. Cells lacking only functional myosin II filaments (early wild-type or *myoII*-null cells) also have multiple Ras-GTP patches approximately symmetric around the cell. However, these cells do extend a protrusion by which one of the Ras-GTP patches in the protrusion is stronger than the others (see also the kymograph of early wild-type in Figure 1c,d). With myosin II filaments, and thereby an active contractile cortex, the starved wild-type cell gets a relatively stable front to rear axis: protrusions appear predominantly in the front. Therefore, these cells adopt reflection symmetry along the front-to-rear axis. Cells do make Ras-GTP patches at the side and in the rear, but usually these patches are not strong enough to overcome the inhibition of the contractile cortex to induce a branched F-actin filled protrusion. In polarized wild-type cells, the strongest Ras-GTP patches and associated protrusions appear in the front alternatingly to the right and left. Thus, in time, gliding reflection symmetry is detectable, comparable to the footsteps of a walker in the snow; two objects with gliding reflection symmetry are identical after one object is glided back opposite the other object and then reflected. A mutant expressing a phospho-mimic form of Scar (ScarS55D [30]) lacks this form of symmetry: Ras patches and pseudopods still appear in the front, but not alternatingly to the right and left, suggesting that Scar or a downstream component mediates the breaking of reflection symmetry into gliding reflection symmetry [4,7].

The experimenter can further intervene with these symmetry forms to induce specific responses; for instance the expression of dominant active Rap1G12V in polarized cells leads to massive destabilization of the contractile cortex (thereby losing longitudinal symmetry) and uniform activation of Ras-GTP (thereby losing rotational symmetry). These cells extend multiple pseudopods in all directions, and since Rap1-GTP induces strong adhesion of the cell to the substratum, these cells get a pancake-like appearance [6,28].

In summary, the Ras/cytoskeleton system can adopt a series of symmetry forms of activated Ras-GTP depending on the presence of different elements of the cytoskeleton. In wild-type cells (Figure 2A–D) the basis is rotational symmetry of Ras-GTP in unpolarized cells (A,B), converting to reflection symmetry of polarized cells (C), and finally to gliding reflection symmetry in polar wild-type cells with positional memory (D). And the coupled excitable Ras/F-actin system then induces a pseudopod at the position of the strongest Ras-GTP patch.

## 6. Kinetic Fine-Tuning of Pseudopod Formation and Symmetry Breaking

Each time the cell extends a pseudopod the symmetry gets broken in a new way. Therefore the principles underlying symmetry breaking are relevant for the kinetics of pseudopod formation and *vice versa*. Detailed kinetics of pseudopod formation revealed that a cell without pseudopod has a stochastic probability of 15%/s to extend a pseudopod. In wild-type cells the probability to extend a second pseudopod is strongly inhibited 3.5 fold in the entire cell, and the probability to extend a third pseudopod is inhibited even 13-fold [6]. Similar complex kinetics of pseudopod extension was also observed in neutrophils, mesenchymal stem cells and the fungus *B.d. chytrid* [6]. In *Dictyostelium* mutants *forAEH*-null and *racE*-null the start of a new pseudopod is inhibited only 1.2 fold by an extending pseudopod, demonstrating that inhibition requires the contractile cortex [6,16]. The contractile cortex generates the longitudinal axis of polarity. This suggests that the extending pseudopod strongly enhances this longitudinal axis of reflection symmetry. As a consequence, cells generally extend only one pseudopod at a time, not zero (then the probability to start a pseudopod is high) and not more than one (then the probability is strongly inhibited).

## 7. Memory and Symmetry Breaking

The extension of pseudopods is also non-random in space. Cells have persistent movement, meaning that new pseudopods are extended in a similar direction as previous pseudopods. Since pseudopods are extended perpendicular to the cell surface, the tendency to move in the same direction implies that new pseudopods start nearby previous pseudopods [2,31,32,33,34]. Somehow cells have a memory of the place in the cell where they extended previous pseudopod(s). Detailed analysis uncovered two types of memory: a long term memory of a polarity axis related to longitudinal front to rear reflection symmetry, and short term memory of position related to the alternating right/left gliding reflection symmetry. The long term memory stores the global position of the last ~11 pseudopods forming a polarity axis from front to rear with a contractile cortex in the rear half of the cell [3,7,35]. New pseudopods are preferentially made in the front 30% of the cell, which provides both persistence of movement and strengthens the polarity axis. Myosin II filaments in the contractile cortex are essential to establish a polarity axis. It is evident that longitudinal symmetry breaking generating reflection symmetry forms the basis for the memory of the polarity axis. Therefore, it appears that longitudinal symmetry breaking leads to a rather stable front-rear polarized cell.

The short term memory of position remembers only the position of the last previous pseudopod, which increases the probability to start the next-next pseudopod at that position [2,5,7,31]. This memory generates two series of pseudopods: odd pseudopods (1,3,5 etc.) starting at the same position and even pseudopods (2, 4, 6 etc.) all starting from another position. In polarized cells with their front-rear polarity axis, these two positions are both in the front 30% of the cell and cells move by alternating right/left pseudopods leading to persistent zig-zag trajectories [2]. A mutant expressing a phospho-mimick form of Scar (ScarS55D) has a polarity axis but no memory of position and therefore extend pseudopods somewhere in the front 30% of the cell but not in alternating right/left order [5,7]. It is evident that gliding reflection symmetry breaking generating excitable hotspots and is the basis for the memory of the position. Therefore, it appears that gliding reflection symmetry breaking is very dynamic with a life-time of only one pseudopod.

## 8. Conclusions

Symmetry in biology is approximate and not exact as in physics. Two aspects of symmetry and symmetry breaking are important during cell movement. Firstly, the symmetry forms are metastable: in rotational symmetry unpolarized cells may have a number of Ras-GTP patches that are approximately equally distributed around the cell. The number of patches may increase or decrease, but the cell retains rotational symmetry. Secondly, transitions to a more complex symmetry are transient and dynamic: an unpolarized cell with rotational symmetry of Ras-GTP patches may start a pseudopod at one of these patches thereby getting an elongated shape with reflection symmetry; when the pseudopod stops, the cell returns to a round shape, but still has rotational symmetry of Ras-GTP patches, and later a pseudopod may start from another Ras-GTP patch. Understanding the molecular basis how rotational symmetry of Ras-GTP patches is modified by cytoskeleton and signaling pathways is instrumental for our understanding how cells can use these symmetry forms to generate specific locomotion behavior (Figure 3). The coupled excitable Ras-GTP/F-actin system triggers at the strongest Ras-GTP patch the nucleation and further polymerization of branched F-actin, leading to the extension of a pseudopod. The extending pseudopod activates two systems: first, the local modification of presumably Scar providing a short term memory that later primes the Ras-GTP/F-actin excitatory to start a new pseudopod at that position. The second system activated by the extending pseudopod is a combination of the global induction of myosin II filaments in the entire cell and its inhibition at the place of the extending pseudopod. In *Dictyostelium* global activation myosin filament formation is mediated by the rapidly diffusing cGMP, produced by a guanylyl cyclase that is activated in the extending pseudopod [7,22], while local inhibition is mediated by Rap1-GTP that is activated in the pseudopod by Ras-GTP [36]. As a consequence of global activation and local inhibition, myosin II filaments are formed in the rear, thereby inhibiting pseudopod formation in the rear and providing a longitudinal symmetry axis. Since this longitudinal symmetry form is relatively stable, it leads to a long-term memory of pseudopods formation in the front of the cell, which is the fundament of intrinsic persistent movement for efficient food searching [2,37] and chemotaxis [38]. The scheme depicted in Figure 3 for *Dictyostelium* probably also hold for other organisms, suggested by comparing kinetics and memory of cell movement in *Dictyostelium*, the fast moving neutrophils, the slow moving mesenchymal stem cells or the fungus *B.d. chytrid* [6,7,39]. Although the molecules and regulatory mechanisms may be different in these organisms, such as the role of Ras versus CDC42 for actin polymerization [40] or cGMP versus Rho-Kinase for myosin polymerization [41], the fundaments of symmetry and symmetry breaking for cell movement may be conserved.

## Figures and Tables

**Figure 1 cells-09-01809-f001:**
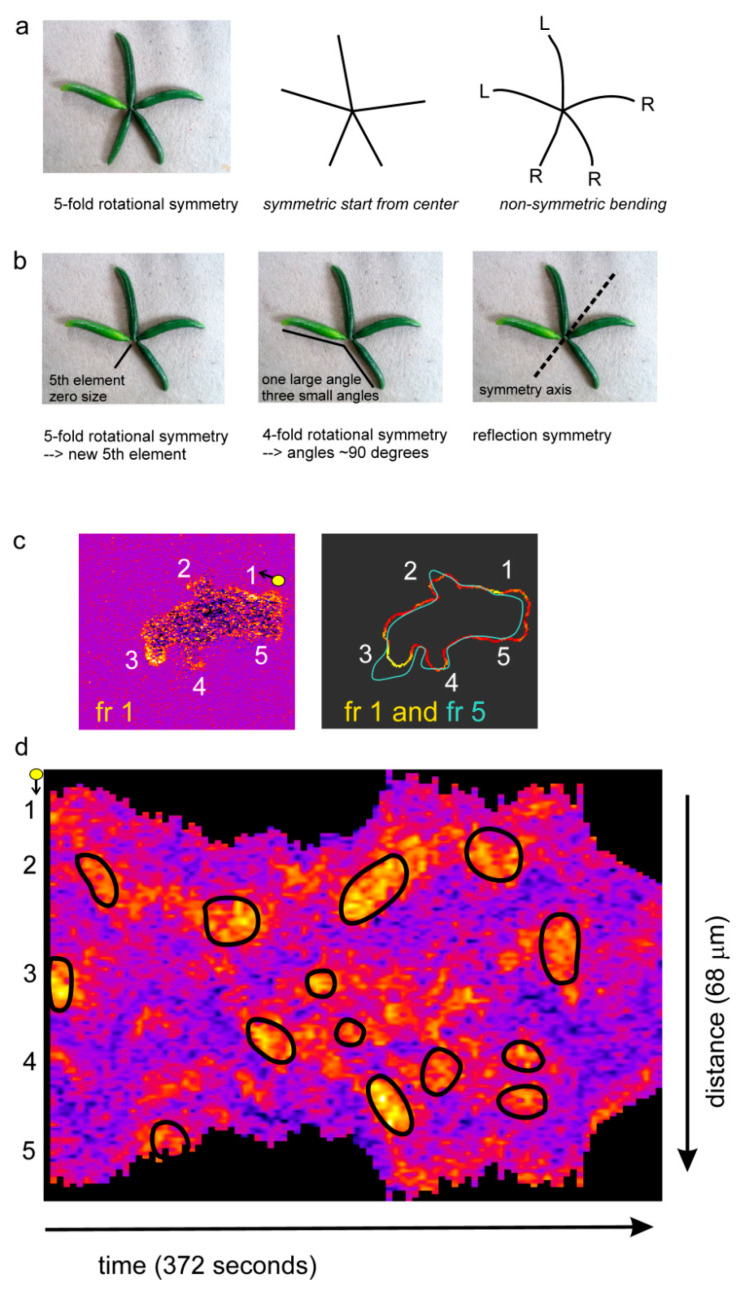
Fundaments of symmetry and dynamic symmetry breaking. (**a**) Symmetry in biology may depend on the viewpoint. The image is an object with 5 elements. The drawing in the middle connects the center of the object with the ends of the elements, showing approximately 5-fold rotational symmetry. The drawing on the right follows the curvature of the elements, showing that the object has poor 5-fold symmetry. (**b**) Distortion of symmetry. Removing one element leads to poor 4-fold and poor 5-fold rotational symmetry, but to very good reflection symmetry. Symmetry can recover in different ways, depending on the underlying molecular mechanism. (**c**,**d**) Unpolarized early *Dictyostelium* cells expressing RBD-Raf-GFP and cytosolic-RFP were followed in time at 4 s per frame, providing a very sensitive sensor for active Ras-GTP. The intensity at the boundary of the cell was measured and is presented in the kymograph. (**c**) Image of frame 1 reveals multiple Ras patches with approximately 5-fold rotational symmetry; the outline of the cell in frame 5 reveals that a pseudopod was extended at Ras-GTP patch 3. In frame 6 a pseudopod will start in patch 2. (**d**) The kymograph reveals about 53 Ras-GTP patches and 14 extending pseudopods (indicated by the back circles); (**c**,**d**) are redrawn from [4].

**Figure 2 cells-09-01809-f002:**
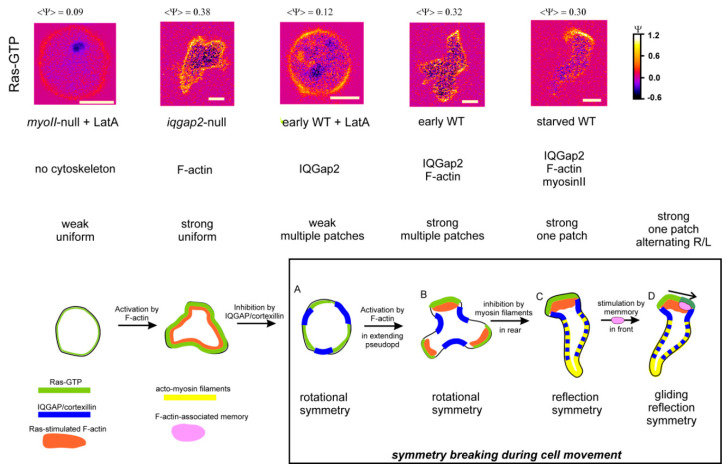
Symmetry and symmetry breaking of Ras-GTP localization in a series of mutants in the absence and presence of the F-actin inhibitor LatA. The mutants are ordered so they form a sequence of symmetry breaking. Top shows images of representative cells. <Ψ> is the average fluorescence intensity at the boundary of the cell (see [4] for definition). Bottom shows schematics with the localization of key components to establish the different forms of symmetry. The box represents the symmetry forms and transitions in wild type *Dictyostelium* cells. The figure is redrawn from [5].

**Figure 3 cells-09-01809-f003:**
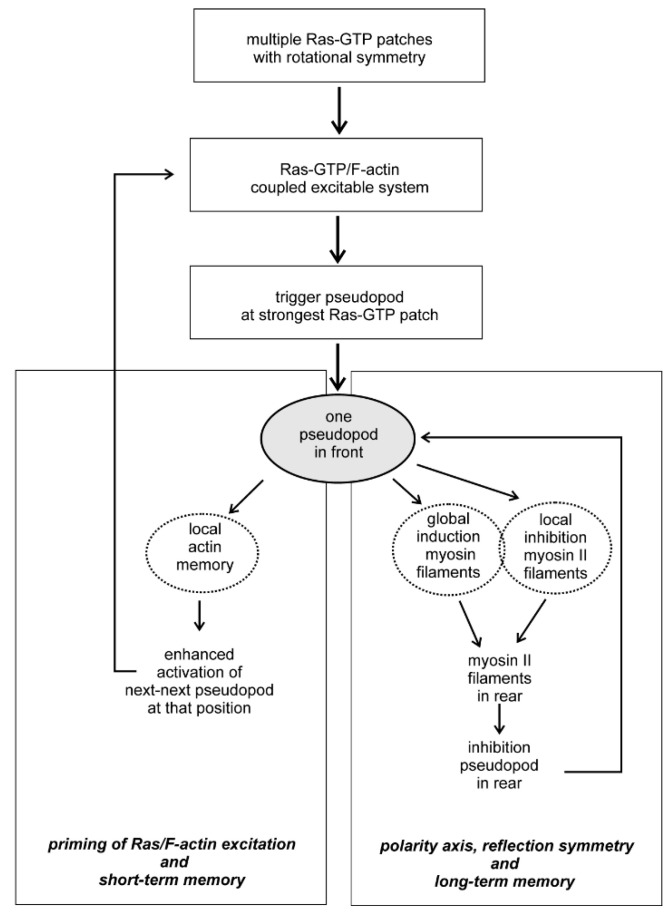
Flow diagram of symmetry, excitability and memory for cell movement.

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
