# Peer review of "Symmetry Breaking during Cell Movement in the Context of Excitability, Kinetic Fine-Tuning and Memory of Pseudopod Formation"

_cells, 2020, doi:10.3390/cells9081809_

Round 1

Reviewer 1 Report

This mostly reads very well.  I largely agree with the views, and I'm surprised no one has fully articulated things like this before.  Mechanisms of cell motility is slightly outside my comfort zone these days, however, for my tastes, I would probably place less emphasis on individual molecules- there is talk about Ras-GTP, but how many Ras proteins does Dicty have?  Ras is not acting alone, so I would caveat this early on, that Ras (which Ras?) seems like a reasonably effective cue for the protrusion, so you will use this as the molecular anchor to your argument from there on.   I think the cortex everywhere in else in the cell is a big part of the persistence (as mentioned in the review), but it is not just Ras and actin that are already there in abundance at the front.  It is everything they normally associate with.

The review is very fluently written, although there are a few minor typos.  I was a bit confused by "The lowest level of symmetry of a cell is a fully symmetric sphere".  This was not my understanding of basic geometry- if you do mean this, it might be good to explain it a little more.

Some other citations that might be good to add.  I think Insall's NCB paper (PMID: 17220879) should probably be mentioned.  As much of the discussion is about "memory", it might be worth talking about other inferred aspects of memory over similar timescales during chemotaxis to cAMP.  For example: PMID: 25249632 and PMID: 24388853, although the latter is rather self-serving as far as the reviewer concerned- and both are talking about persistent behaviour to cAMP, rather than any kind of intrinsic memory- which might be a contrast to think about.  Along these lines, can you contextualise the function of intrinsic persistence to the biology?

Author Response

I am grateful to the reviewers who made insightful comments that allowed me to improve the manuscript, and especially made several points on symmetry more clearly. The changes made are indicated in red in the revised manuscript.

Reviewer 1.

point 1. The arguments to put emphasis on Ras as marker of symmetry are given in lines 43-47 of the revised manuscript.

point 2. I agree with the reviewer that the contractile cortex is a major component of the persistence. This is explained more clearly in line 146 of the revised manuscript.

Point 3. In response to the remarks of both reviewers, they are very correct that the sections on the order of symmetry is confusing. Actually it was not correct in the original manuscript. The term order is confusing because in symmetry it has a very different meaning than in thermodynamics. As the symmetry breaks from rotational symmetry to reflection symmetry, the system becomes more ordered thermodynamically, while the order of symmetry (the number of viewpoints) decreases. In the revised manuscript I have indicated  in lines 60-66 the potential confusion when using order, and indicated that in the manuscript symmetry transitions are described as less or more complex symmetry forms.

Point 4. I have added a reference to the paper of Andrews and Insall (PMID: 17220879), which actually was the trigger to study pseudopods and persistent cell movement. Reference to PMID: 25249632 was made in the original manuscript as reference 33 (now reference 35). I have not made reference to PMID: 24388853 in the revised manuscript. This very nice paper is on transcriptional regulation; I find it difficult to connect the observations to symmetry of movement.

Point 5. The function of intrinsic persistence to the biology has been described in lines 291-292 of the revised manuscript.

Reviewer 2 Report

This manuscript for the essay article entitled "Symmetry breaking during cell movement in the context of excitability, kinetic fine-tuning and memory of pseudopod formation" discusses the author's view on how series of symmetry breakings leads to the formation of cell movement based on the experiment of Dictyostelium cells. I find it interesting and unique to discuss the cell movement from the viewpoint of series of symmetry breaking. I support publication of this manuscript after the author address the following point. 

  1. A symmetry breaking implies that the degree of symmetry is reduced.  For instance, comparing circles and pentagons, the symmetry is higher for circular shapes than for pentagonal shapes. In this manuscript, the author argues that symmetry breaking leads to a state with "higher order" or "higher order symmetry". This expression is confusing. For instance, the author aruges that "the high order symmetry state may depend on a symmetry state with lower order. " It is confusing as to which one depends on which state. The author should take into account the difference between "high" and "low" in the degree of symmetry. 
  2. In line 133, it is not clear in what respect bF-actin and pF-action have opposite function. 
  3. In line 149, the reason of why only one psudopod is formed and it is formed in the front is not clear to me. Is there a possiblility that  some lateral inhibition process may restrict the muptiple formation of extensing pseudopod and inhibit a strong Ras activation in the lateral region.  
  4. In lines from 190 to 194, I think a time translational symmetry implies that the behavior is uniform in time. For instance, it is broken when the system exhibits an oscillatory behavior. Thefore, I think the alternating appearance of left and right protrusion may indicate the time translational symmetry breaking. Also lines from 249 to 252, the discussion is difficult to understand.
  5. Whereas the author claims that the model of cell movement presented in this manuscript is conserved in different organisims, I think comparison between Dictyostelium cell movement and other organisms is limited and not sufficient to convince. The authors should have a more in-depth discussion on this aspect. 

Author Response

I am grateful to the reviewers who made insightful comments that allowed me to improve the manuscript, and especially made several points on symmetry more clearly. The changes made are indicated in red in the revised manuscript.

Reviewer 2.

point 1. In response to the remarks of both reviewers, they are very correct that the sections on the order of symmetry is confusing. Actually it was not correct in the original manuscript. The term order is confusing because in symmetry it has a very different meaning than in thermodynamics. As the symmetry breaks from rotational symmetry to reflection symmetry, the system becomes more ordered thermodynamically, while the order of symmetry (the number of viewpoints) decreases. In the revised manuscript I have indicated  in lines 60-66 the potential confusion when using order, and indicated that in the manuscript symmetry transitions are described as less or more complex symmetry forms.

Point 2. The opposite functions of BF-actin and pF-actin have been explained better in lines 146 of the revised manuscript.

Point 3. The extending pseudopod appears to inhibit the start of a second pseudopod about 3.5 fold. This inhibition requires the contractile cortex (as persistence), but inhibition occurs in the entire cell. This has been explained added in line 230 of the revised manuscript.

Point 4. Translational symmetry is not the best way to explain the observations. In the revised manuscript I have changed this to gliding reflection symmetry, and explained it better, mainly by adding more information at lines 199-202 of the revised manuscript.

Point 5. Additional information is provided in the revised manuscript on the aspect of pseudopod formation that is identical between these four organisms in lines 231-232 of the revised manuscript.